# Un-Doubling Diffusion: LLM-guided Disambiguation of Homonym Duplication

## Abstract

Homonyms are words with identical spelling but distinct meanings, which pose challenges for many generative models. When a homonym appears in a prompt, diffusion models may generate multiple senses of the word simultaneously, which is known as homonym duplication. This issue is further complicated by an Anglocentric bias, which includes an additional translation step before the text-to-image model pipeline. As a result, even words that are not homonymous in the original language may become homonyms and lose their original meaning after translation into English. In this paper, we introduce a method for measuring duplication rates and conduct evaluations of different diffusion models using both automatic evaluation utilizing Vision-Language Models (VLM) and human evaluation. Additionally, we investigate methods to mitigate the homonym duplication problem through prompt expansion, demonstrating that this approach also effectively reduces duplication related to Anglocentric bias. The code for the automatic evaluation pipeline is publicly available.

## 1 Introduction

In recent years, diffusion models Ho et al. (2020) have made remarkable progress in the field of image generation; however, they still face challenges in accurately mapping text to images, especially in cases of lexical ambiguity. It occurs when a single word or phrase has multiple meanings, resulting in uncertainty or multiple possible interpretations within a given concise context. A specific instance of lexical ambiguity is homonyms, words that have multiple distinct, unrelated meanings (e.g., "palm" referring to the part of the hand or a type of tree). While humans typically resolve such ambiguities using real-world information, diffusion models often lack access to extended context.

Human communication adheres to the single-meaning-per-symbol axiom Rassin et al. (2022), whereby each word in a sentence conveys only one specific meaning and there can be no other. However, as noted in several recent studies Rassin et al. (2022); White & Cotterell (2022), diffusion models exhibit behavior inconsistent with this principle: a single word can be interpreted as two entities (see examples in fig. 1). When a homonym appears in a prompt, in an attempt to satisfy all possible variants of the word, diffusion models adopt a precautionary strategy and generate multiple possible senses within a single image (i.e., duplication of the homonym is observed). This behavior is attributed to the way CLIP (Contrastive Language–Image Pretraining) Radford et al. (2021) represents homonyms: it encodes each word as a linear superposition of their different meanings White & Cotterell (2022).

This problem is further compounded by the prevalence of English data in training sets of image generation models. Such an anglocentric bias results in the homonym duplication, even in cases where the homonym is not present in the original language of the prompt. For example, the Russian non-homonymous word "свидание" (meaning social meeting) translates to the English homonym "date", which can cause unintended image generations of either the fruit or a calendar date. This behaviour occurs because English is used as the anchor language, and the text encoder processes only the translated prompt, where an original unambiguous word may become a homonym.

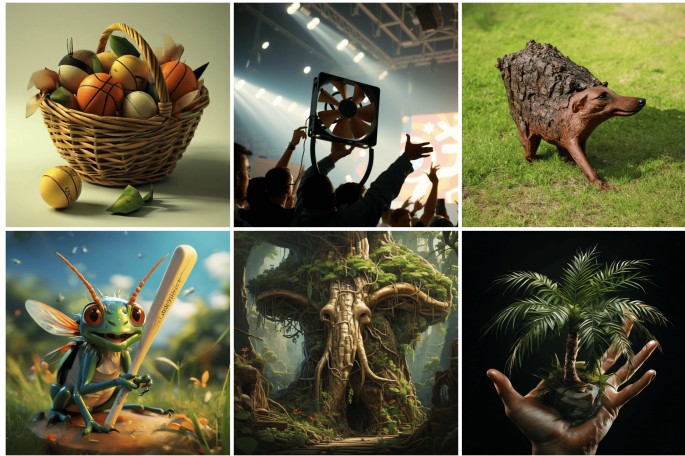

Figure 1: Homonym duplication examples. Words in the top row (from left to right): "basket", "fan", "bark". Words in the bottom row: "cricket", "trunk", "palm".

This study aims to quantify the frequency of homonym duplication in diffusion models. We introduce two evaluation methods: automatic ranking with VLM-like and CLIP-like models, and human evaluation via crowdsourcing. Eleven diffusion models are assessed using a novel multimodal homonym benchmark. Additionally, we explore prompt expansion guided by large language models to mitigate homonym duplication, including translation-induced cases. Our source code for automatic evaluation and prompt expansion is publicly available at `https://anonymous.4open.science/r/Un-Doubling-Diffusion-662E/`.

Our contributions can be summarized as follows:

- We propose a Human Evaluation (HE) pipeline to measure the frequency of homonym duplication, and use it to quantify duplication rates in several diffusion models.
- A benchmark of homonyms with their English and Russian senses has been compiled and released as open-source. While this study focuses on English and Russian, the findings may be extended to other languages.
- We perform VLM-based Automatic Evaluation (AE), while also conducting a comparative analysis between automatic and human evaluation methods. The source code is publicly available.
- This study provides the first quantitative evidence that LLM-based prompt expansion reduces duplication rates, including translation-related homonym duplication.

## 2 RELATED WORK

**Homonym duplication in diffusion models.** Numerous studies have investigated the phenomenon of polysemous words in natural language processing and computer vision, focusing on how these words are represented within models and the behaviors they elicit. Arora et al. (2018) demonstrated that the various meanings of polysemous words are encoded as a linear superposition within the embedding of the word. Consequently, the duplication of homonyms observed in generative models can be attributed to the polysemy that is inherently present in embedding spaces. Rassin et al. (2022) conducted the first study devoted to the problem of homonym duplication in diffusion models (specifically in DALLE-2 Ramesh et al. (2022)). The authors utilize a specialized contextual prompt to trigger multi-sense generation and achieve ambiguity in the generated results. However, this work only focuses on DALLE-2 and does not examine other diffusion models. White & Cotterell (2022) introduces the term "superposition of homonyms" in the context of image generation with diffusion models. This term refers to the tendency of diffusion models to simultaneously

generate visual representations corresponding to all possible senses of a homonym until sufficient disambiguating context is provided.

In addition to describing the problem, several studies have focused on developing methods to address homonym duplication. For example, Lee (2021) proposes an approach that involves detecting homonyms in text using word embeddings and replacing them with synonymous words that are not homonyms, thus reducing ambiguity. Mehrabi et al. (2022) propose to use an additional filter language model during generation to determine user intent. The model can ask clarifying questions or produce multiple candidate outputs simultaneously. Furthermore, the authors introduce a benchmark designed to evaluate the effectiveness of disambiguation following user feedback. It is important to note that their work addresses the broader issue of lexical ambiguity rather than focusing specifically on homonyms, which is the primary focus of this paper. The previously mentioned White & Cotterell (2022) proposes using linear algebra techniques to shift the homonym embedding to the desired meaning.

**Anglocentrism in generation models.** Models are trained predominantly on English data; consequently, their performance in other languages is lower than in the predominant language (even when the tokenizer accounts for tokens from multiple languages). For example, Xing et al. (2025) shows that Pixart Alpha Chen et al. (2023) has an average CLIPScore Hessel et al. (2022) on non-English prompts that is 9.2 points lower than on English prompts (29.8 vs 39.0), while translating the prompts from the source language into English increases the metric to 38.3 and 39.7 by two different translators. As a result, current methods for non-English image generation generally use a translation-first approach, where non-English prompts are translated into English prior to processing, as stated in Derakhshani et al. (2025). This approach causes semantic drift, where subtle meanings may shift, and originally unambiguous words can become homonyms after English translation.

## 3 Homonym Benchmark

### 3.1 Homonym List Compilation

**LLM Usage.** We employ LLMs to help with data collection and processing. In particular, LLMs are used to (1) compile the initial list of candidate homonym words, (2) to obtain the most common homonym meanings and their corresponding frequency of use. As a first step, 330 homonym words and 765 corresponding meanings (2 to 5 most common meanings per homonym, including both noun and verb senses) are obtained using modern LLMs: DeepSeek-R1[1] DeepSeek-AI et al. (2025) and GPT-4o[2] OpenAI et al. (2024). Models are asked to retrieve a list of homonyms (candidates), along with their senses ranked by frequency of use, accompanied by examples for each sense. After that, models validate each other's candidate lists, and the resulting combined list is sent to experts.

Linguists further validate the compiled list by selecting words based on their frequency of use. After compiling the list of homonyms and their meanings, for each meaning, English and Russian definition is taken from open-source resources and dictionaries such as COCA Davies (2015) and BNC Consortium (2007) corpuses, online dictionaries Cambridge University Press (n.d.); Merriam–Webster (n.d.), as well as English homonym dictionaries Malakhovskiy (1995); Gorulko-Shestopalov (2021).

### 3.2 Validation and Visual-based Aggregation

Further processing and verification of the list is carried out manually by experts with a higher education degree in linguistics. To guarantee the highest quality of the final list, we employ a triple overlap method that adheres to specific criteria:

- **Meaning relevancy.** Preference is given to modern and frequently used meanings. Outdated or highly specialized meanings are excluded.

---

[1]WebUI: chat.deepseek.com; usage window: 22 January–10 February 2025.
[2]WebUI: chatgpt.com; usage window: 22 January–10 February 2025.

- **Feasibility of visual representation.** The final list includes only meanings that can be clearly and unambiguously visualized. For example, the meanings of "well" as a hydraulic structure (visualizable) and as an adverb indicating quality (not visualizable) are excluded. In contrast, the meanings of "mole" as a small mammal and as a dark skin mark (both visualizable) are included.

- **Semantic distinction.** Meanings of a homonym must be distinct, not just variations of the same concept. For example, "cart" can mean a small hand-pushed carrier, a horse-drawn vehicle with two or four wheels, or specifically a two-wheeled horse-drawn vehicle. These related senses can be difficult to distinguish in generated images; therefore, they are excluded from the list.

- **Meanings are not nested within each other.** For instance, the word "orange" can denote both "the fruit of the citrus tree" and "the color between red and yellow". Because oranges are inherently orange in color, it is challenging to separate these meanings distinctly. To address this, we exclude such words from our list.

Based on these criteria, each expert assigns a rating to each meaning according to the following scale: (0) — does not match the criteria (to be excluded from the final list), (1) — partially matches the criteria (to be discussed), (2) — fully matches the criteria (to be included in the final list). In cases of rating discrepancies, a joint discussion is held using the aforementioned online resources and dictionaries. As a result, the final list comprises 171 homonyms, each with its corresponding senses in both English and Russian.

## 3.3 Experts and Roles

Two groups of experts are involved in the comprehensive development of the dataset:

1. **3 linguists** are involved in the creation of the final list of homonyms. The selected experts hold a master's degree in linguistics, possess relevant professional experience, and are familiar with using LLMs.

2. **2 translators** are involved for validation and enrichment of homonym meanings, initially obtained using LLMs. The translators also hold a master's degree in linguistics, as well as over three years of experience in translation.

## 4 Human Evaluation

### 4.1 Image Generation

To estimate duplication frequency, it is necessary to generate images for each homonym that will be evaluated for the simultaneous presence of multiple meanings. We explore the following open-source models: Stable Diffusion 3 (Medium) Esser et al. (2024), Stable Diffusion 3.5 (Medium, Large) Esser et al. (2024), Stable Diffusion XL Podell et al. (2023), Pixart (Alpha, Sigma) Chen et al. (2023; 2024), Kandinsky 3 Arkhipin et al. (2024), Playground 2.5 Li et al. (2024), Flux 1 (schnell, dev) Labs (2024), CogView 4 Zheng et al. (2024).

We utilize the Hugging Face Diffusers framework von Platen et al. (2022) and configure the generation parameters according to the official model specifications. We set the height and width to 1024 pixels for all generations. Seeds are selected from 0 to 49 inclusive, so that 50 generations are performed for each homonym by all models in single-sample inference mode, while maintaining determinism. In total, for all 11 models, we generate $50 \cdot 171 \cdot 11 = 94.050$ images to ensure a reliable evaluation.

### 4.2 Crowdsource Annotation Pipeline

Human evaluations of homonym duplication in the generated images were obtained using the TagMe [3] and Yandex Tasks [4] crowdsourcing platforms. To ensure the reliability of

---

[3] https://tagme.sberdevices.ru

[4] https://tasks.yandex.ru

these evaluations, we implemented training and examination phases, along with several quality control measures, including dynamic overlap aggregation, daily task limits, honeypot tasks, and response-time blocking mechanisms. For more information on the crowdsource annotation pipeline, please see appendix A.1. Overall, the image labeling task involved a pool of 1,436 annotators who collectively completed a total of 438,667 samples. Of the 104,450 images, 94,954 (90%) were successfully aggregated. The crowdsourcing task interface and the annotation statistics can be seen in fig. 2.

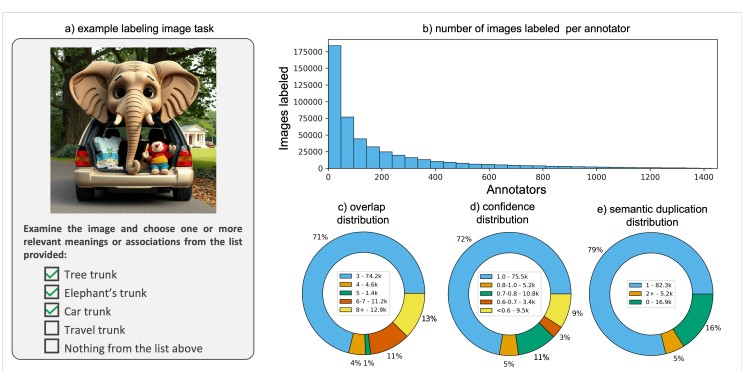

Figure 2: Annotation characteristics and distribution analysis. a) Crowdsourcing task interface for image labeling. b) Number of images labeled per annotator. c) Distribution of annotation overlap. d) Distribution of annotation confidence. e) Distribution of semantic duplication of annotation.

## 5 Automatic Evaluation

Feizi et al. (2025); Yasunaga et al. (2025) demonstrate that the VLM-as-a-judge approach shows great potential for reliable evaluation in vision-language tasks. We experiment with automatic evaluation, and use Qwen2.5-VL Bai et al. (2025) as a judge. We sample $N$ independent sequences per image from the stochastic decoder, parse the binary verdicts, and compute the empirical probability as in 1, using the indicator function defined in 2 to assign decisions to each sequence. We adopt a chain-of-thought prompt to elicit long, step-by-step explanations Zhang et al. (2025). From each sequence of VLM responses, we take the final sentence of the form "DUPLICATE: TRUE" or "DUPLICATE: FALSE" from which the binary answer $v_i$ is parsed by the deterministic parser $\pi$ 3.

$$\widehat{p}(x;\theta) = \frac{1}{N}\sum_{i=1}^{N} r(y_i), \quad i = 1, \ldots, N, \tag{1}$$

$$r(y_i) = \mathbf{1}\{v_i = \mathtt{true}\} \in \{0, 1\}, \tag{2}$$

$$v_i = \pi(y_i), \tag{3}$$

where $y_i$ is the i-th sequence from VLM, $x$ is the prompt (image, text) and $\theta$ is the generation parameters. The overall evaluation pipeline is shown in fig. 3.

We experiment with two setups: one-stage and multi-stage inference prompts. The one-stage prompt directly asks the model if each sense is present in the image. The multi-stage prompt breaks the task into sequential steps, such as listing objects and analyzing the meanings of homonyms. For the one-stage setup, we test different ways of verbalizing homonym meanings: in setup $p_1$, each meaning is given with both a Russian translation and definition; in $p_2$, the translation is in Russian but the definition is in English; and in $p_3$, only the English definition is provided without a Russian translation. In the $p_2$ setting, the second language is employed to help the model effectively disentangle the representations of meanings within the image. Examples of one-stage and multi-stage $p_3$ prompts, as well as the model answer, can be found in appendix A.2.1.

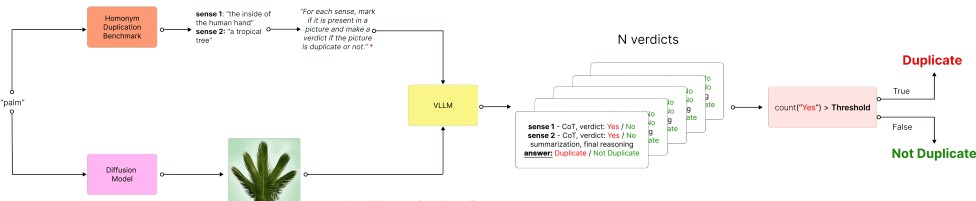

Figure 3: The overall pipeline of automatic evaluation. VLM evaluates images generated for each homonym sense, providing multiple reasoned responses, and images are flagged as duplicates if "duplicate" votes exceed a set threshold.

# 6 RESULTS

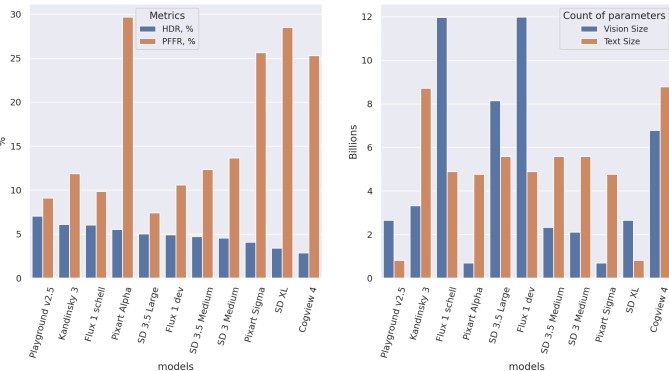

Figure 4: Per-model Homonym Duplication Rate (HDR) and Prompt Following Failure Rate (PFFR) with corresponding model sizes.

We define the Homonym Duplication Rate (HDR) metric 4 as the average duplication percentage of the selected model for each homonym:

$$HDR = \frac{1}{\sum_{i=1}^{H} K_i} \cdot \sum_{i=1}^{H} \sum_{j=1}^{K_i} \mathbf{1}\{m(pic_{i,j}) = \texttt{true}\} \cdot 100\%, \qquad (4)$$

where $H$ is the number of homonyms, $K_i$ is the number of generation seeds ($K_i$ is 50 for all homonyms), $pic_{i,j}$ is the j-th image in the row generated for the i-th homonym and $m$ is either human preferences or model evaluation aggregation, depending on the evaluation type (human or automatic). In the case of human preferences, for stability, we define $m$ as a majority vote over the set of options (i.e., if the most frequent response indicates multiple meanings, the image is classified as a duplicate). In contrast, for VLM evaluation, an image is deemed a duplicate if all $N$ of its chains-of-thought (where $N$ is set to 10) contain a "True" verdict.

## 6.1 HUMAN EVALUATION

The per-model results of human preferences are shown in fig. 4; model sizes are also represented. In addition, we include the Prompt Following Failure Rate (PFFR) metric, which represents the number of cases where workers label the depicted senses as "nothing from the list above", implying that the model does not follow the prompt. As can be noted, Playground 2.5 is the most frequently duplicating model. Cogview 4 duplicates the least, but it can be attributed to the fact that it rarely follows the prompt, considering the PFFR. It is also worth noting that there is no correlation between HDR and the size of the vision and text components of the model.

Table 1: Alignment between homonym evaluation and VLM-based automatic evaluation results. We denote AUROC* with an asterisk (*) to indicate the lack of ground-truth labels in this task. Sense representation type indicates the different ways in which homonym senses can be embedded into a prompt (see section 5).

| Prompt type | Sense representation type | $r \uparrow$ | $\rho \uparrow$ | JSD $\downarrow$ | OPA $\uparrow$ | AUROC* $\uparrow$ |
|---|---|---|---|---|---|---|
| one-stage | $p_1$ | 0.269 | 0.232 | 0.840 | 0.919 | 0.722 |
| | $p_2$ | 0.248 | 0.215 | 0.849 | 0.918 | 0.707 |
| | $p_3$ | 0.265 | 0.232 | 0.840 | 0.920 | 0.718 |
| multi-stage | $p_1$ | 0.369 | 0.338 | 0.790 | 0.918 | 0.830 |

## 6.2 Automatic Evaluation

**VLLM-based Evaluation.** We measure the alignment of VLLM responses with human evaluation in terms of the Pearson correlation coefficient $r$, Jensen–Shannon divergence (JSD), Spearman's rank correlation coefficient $\rho$, AUROC*, and Overall Percent Agreement (the percentage of total samples for which the two methods produce the same binary outcome). The results can be seen in table 1. Human evaluation results cannot be considered ground truth due to task complexity, as one in ten images lacked consensus among crowd workers (section 4). Nevertheless, to assess the alignment between human and automatic evaluations, we compute AUROC, treating human labels as the ground truth, denoted as AUROC* to highlight this distinction. Despite low correlation coefficients, the overall percent agreement (OPA) is high due to class imbalance, as 95% of images are labeled as non-duplicates according to human evaluation (fig. 2(e)).

**Ablation Study on different sense representation types.** We calculate the alignment metrics between automatic evaluation results with different sense verbalization types in the prompt, as described in section 5. The results are shown in table 2. The correlation between the metrics is moderate overall: a relatively strong correlation is observed between $p_1$ and $p_2$, while the correlations between $p_1$ and $p_3$ and between $p_2$ and $p_3$ are weaker. The JSD values for these comparisons are below the moderate threshold of 0.5.

**CLIP-based Evaluation.** Additionally, we assess three CLIP-like rankers as a tool for the automatic evaluation of homonym duplication and compare the obtained metric with human evaluation results. We utilize two multilingual SigLIP models Zhai et al. (2023); Tschannen et al. (2025) as well as the OpenAI CLIP L-14 model Radford et al. (2021). One can observe a negligible correlation between CLIPScores and human judgments in terms of correlations (see table 4 in the Appendix). Across models, the highest AUROC* occurs with top-2 (second-highest CLIPScore), matching one-stage VLM inference but falling short of multi-stage. This discrepancy stems from CLIP's limited ability to handle cases where meanings are linked through associations.

## 6.3 Proper Name Bias

In certain cases, the model demonstrates a bias toward proper names. For instance, when given the word "stitch", the model frequently produces the cartoon character named Stitch. Similarly, for the word "bat", it often generates the character Batman, even though the words "bat" and "Batman" are spelled differently. In the Appendix, table 5 presents several examples comparing the frequency of proper name generation relative to other meanings; the HDR metrics are obtained through human evaluation for all 11 diffusion models. Generations depicting this bias can be seen in fig. 7.

## 7 LLM-based prompt expansion

Studies show that techniques such as prompt beautification Arkhipkin et al. (2024) and prompt expansion Datta et al. (2023) enhance image aesthetics and diversity. We aim

Table 2: Ablation study of the correlation between automatic evaluation for different sense representation types in the prompt.

| Sense representation type | $r \uparrow$ | $\rho \uparrow$ | JSD $\downarrow$ | OPA $\uparrow$ |
|---|---|---|---|---|
| $(p_1, p_2)$ | 0.829 | 0.766 | 0.388 | 0.802 |
| $(p_1, p_3)$ | 0.731 | 0.694 | 0.481 | 0.786 |
| $(p_2, p_3)$ | 0.722 | 0.693 | 0.485 | 0.784 |

to demonstrate that using a pretrained LLM to expand single-word ambiguous prompts lowers duplication rates in diffusion-based generation. We utilize the compiled homonym benchmark (see section 3) and, for each of 171 words, iterate the seed from 0 to 49 to generate expanded text sequences with the LLM, which are then used as prompts for the diffusion model. We intend to demonstrate a working proof of concept using a single Pixart Alpha model, rather than replicating the demonstration across all models, which would double the annotation effort.

We prompt Qwen3-A3B-30B Yang et al. (2025) model to write an expanded text-to-image generation prompt for each homonym word, and measure the resulting HDR. Specifically, we calculate the count of duplicates over 50 generations for each homonym and then aggregate these rates across all homonyms. We compute the HDR using human evaluation and automatic evaluation. According to human evaluation, the HDR metric scores are 5.54 before prompt expansion and 5.03 ($-9.2\%$) after, while automatic evaluation yielded scores of 9.58 and 5.66 ($-41\%$), respectively. One can observe a decrease in HDR after the prompt expansion, regardless of the evaluation method, indicating that LLM-based prompt expansion can effectively alleviate the duplication problem.

## 8 Anglocentrism as a Related Problem

To study Anglocentrism related to homonym duplication, we simulate a pipeline generating images from short, unambiguous non-English prompts (in Russian). Our primary goal is to determine the frequency of unintended or duplicated meanings. For the experiment, we utilize homonyms collected from our benchmark along with their corresponding short Russian translations. To ensure a valid comparison, we apply the following criteria: (1) homonyms that include at least one verb sense are excluded, as single-word verbs are less likely to be used as prompts; (2) all English translations of homonyms are verified to be consistent with the Russian source through back-translation. Specifically, the madlad-7b translator Kudugunta et al. (2023) in Russian-English mode is used to obtain the English homonyms. After following these steps, we obtain 37 senses of 17 homonym words that have a bipartite English-Russian matching: each meaning's English translation reversely translates into the same Russian word, establishing a bidirectional one-to-one mapping between their meanings across languages. We expand the prompt using a method similar to that described in section 7, with the expansion applied to the Russian input text before the translation.

For translated original and expanded prompts, images are generated by the Playground 2.5 model. An illustration of the prompt expansion pipeline for a non-English prompt is provided in fig. 5. To measure the effect of prompt expansion, we calculate two metrics: the homonym duplication rate (note that homonyms appear in the English translation) and the wrong sense rate (WSR). The WSR represents the proportion of instances where the model generates images reflecting an unintended homonymous meaning rather than the one intended by the user. The results are presented in table 7 in the Appendix. The average WSR decreases significantly after prompt expansion, dropping from 50% to 22%. That is, without prompt expansion techniques, a non-English-speaking user encounters an alternative (unrequested) sense in 50% of generations. Prompt expansion in the source language before translation improves the situation significantly. Concurrently, the HDR also reduces from an

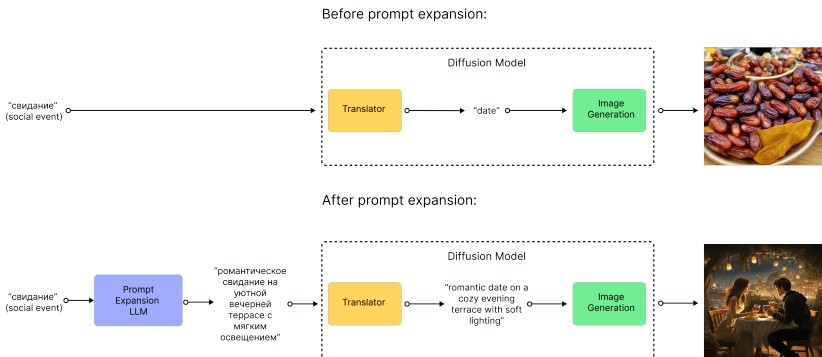

Figure 5: The example of a prompt expansion pipeline for non-English prompts (in this case, Russian) to avoid homonym duplication and sense entanglement caused by translation.

average of 16.5% to 8.9%. These results indicate that prompt expansion effectively mitigates issues related to homonym duplication that occur when translating into English.

## 9 LIMITATIONS AND FUTURE WORK

**Perception bias.** Identifying duplicates is a complex task heavily influenced by individual perceptions and associations. It is not always possible to make definitive judgments for all images. To simplify the evaluation, certain words and their specific meanings, as described in section 3.2, were excluded from consideration. This approach diminishes the uncertainty but does not eliminate it. The examples of easy and complex cases can be found in fig. 6 in the Appendix.

**Absence of sense frequencies.** Another limitation of our study is that the selection of individual homonym senses mentioned in section 3.1 is based on approximate frequency estimates due to the absence of publicly available statistical data. A possible direction for future research would be to analyze English language corpora to determine the actual frequency of each sense for homonymous words.

**Alternative image generation methods.** In this work, we focus specifically on the issue of homonym duplication in diffusion models, excluding other image generation approaches such as autoregressive models (e.g., Tian et al. (2024)) from our scope. The behavior of these models when processing homonyms in prompts may differ substantially and could require alternative solutions, representing a valuable direction for future research.

**Sensitive content.** To avoid unintentional distribution of potentially unacceptable (NSFW) material, we do not publish the generated images. Since single-word homonyms involve few tokens, models, which are trained on average token counts of 15–18 Wu et al. (2024); Byeon et al. (2022), may still output sensitive content. Prior work Betker et al.; Chen et al. (2024); Esser et al. (2024) confirms that training on long synthetic descriptions improves metrics but worsens out-of-distribution issues when inferring from few tokens, supporting our concern.

## 10 CONCLUSION

This paper addresses the challenge of homonym duplication in diffusion models. Our proposed benchmark and comprehensive evaluations provide a systematic framework for quantifying duplication rates across different models. To the best of our knowledge, this is the first study to investigate the homonym duplication problem in the context of the Anglocentric bias in image generation models. We also demonstrate that prompt expansion effectively reduces duplication, including translation-related cases. These findings contribute valuable insights toward improving the reliability of text-to-image generation systems, and the publicly available evaluation pipeline offers a practical tool for future research in this area.

## 11 Ethics Statement

Certain words were excluded from consideration due to ethical concerns. For instance, the word "race" often leads models to reproduce racial biases by generating images of people of color in racing attire. Detecting duplicates in such cases is challenging without perpetuating these biases. Therefore, the word "race" was omitted from our benchmark.

All crowd workers participating in the benchmark creation were fairly compensated. Since homonym duplication labeling is non-trivial and heavily influenced by individual associations, workers were still paid even if they were blocked after making an error in the verification honeypot task (see appendix A.1 for more information).

## 12 Reproducibility Statement

To perform VLM-based evaluation, we use the vllm framework Kwon et al. (2023) version 0.10.0. Even when employing greedy decoding with a temperature of 0 and fixing the seed, strict determinism is not guaranteed by the official vllm documentation[5]. To address this limitation and enhance the reliability of the metrics, we generate and evaluate $N$ sequences per image, as described in section 5. For all generation tasks (including image generation and prompt expansion in both English and Russian) we set seeds ranging from 0 to 49 inclusive to ensure complete determinism. It is important to note that, for prompt expansion, the seed used to generate each expanded prompt is recorded and subsequently applied to generate the corresponding image within the original pipeline. To ensure reproducibility, we provide the complete source code for all stages of this work, including VLM evaluation, image generation, prompt expansion, and metric calculation, as well as the specifications for the conda environment requirements at `https://anonymous.4open.science/r/Un-Doubling-Diffusion-662E/`.

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

# A  Appendix

## A.1  Crowdsource Annotation Pipeline

As stated in section 4, we utilized crowdsourcing platforms to perform human evaluation. A key advantage of the crowdsourcing approach is its capacity to gather annotations from a demographically and professionally heterogeneous group of participants, mitigating potential biases inherent in homogeneous annotator pools. The task instructions required crowdworkers to view an image and select all applicable associations and semantic duplications from a provided list, or to indicate that the image contained no such associations.

### A.1.1  Participants selections.

A two-stage system involving training and exam tasks is used to select crowdworkers. The training and exam tasks are based on pre-defined, unambiguous correct answers.

**Training.** The purpose of the training phase is to screen crowdworkers for their ability to understand the instructions and navigate the task interface. The training phase consists of five tasks. A total of 7,765 crowdworkers began the training phase, but only 6,477 correctly completed at least three tasks, meeting our 50% accuracy threshold.

**Exam.** Crowdworkers who achieve a score of more than 50% correct answers in the training phase are admitted to the qualification exam. The exam consists of 21 tasks. As in the training phase, these tasks feature pre-defined gold-standard tasks with unambiguous correct answers. A total of 6,477 crowdworkers began the exam phase; 5,297 completed all 21 tasks, but only 1,138 correctly completed at least 18 tasks, meeting our 85% accuracy threshold.

### A.1.2  Images annotation.

Access to the main annotation tasks is granted only to crowdworkers who achieve an exam accuracy score of at least 85%. Since we have not screened the images for potentially NSFW content (see more details in section 9), the task pool requires participants to be 18 years or older. To enhance the quality of image annotation, three safeguard mechanisms are implemented: (1) rapid responses — annotators who label images too quickly (in less than 1 second) are temporarily blocked for 14 days+, (2) daily task limit — each annotator is assigned no more than 200 tasks per day to ensure user heterogeneity, (3) honeypot tasks — main tasks are interspersed with honeypot tasks (pre-annotated items with known correct answers), incorrect responses to honeypot tasks lead to the annotator's block. Honeypot tasks are introduced with a 10% probability per task assignment. In total, the unique set of honeypot images accounted for roughly 2% ($\sim$ 2000 images) of the dataset.

### A.1.3  Labeling aggregation.

For reliable image annotation, we apply a dynamic overlap approach. The initial overlap for each image is set to 3, i.e., each image is independently annotated by at least three different annotators. If the required response agreement is below 0.7, the overlap is increased until the desired agreement level is reached. The maximum overlap is set to 9; for overlaps of 8 and 9, the response agreement threshold is lowered to 0.6. The responses are considered consistent

only if they match exactly: both the specific associations selected from the proposed list and the number of associations provided. Images that do not reach the required agreement, even with 9 responses, are labeled as "not_aggregated".

### A.1.4 CHARACTERISTICS.

The image labeling task involved annotators ranging in age from 18 to 94 years, who collectively completed a total of 438,667 assignments and 57,822 honeypot tasks. Of the 104,450 images, 94,954 were successfully aggregated. The required inter-annotator agreement could not be achieved for 9,496 images. On average, annotators spent 3.8 seconds labeling each image. A total of 455 annotators were disqualified as a result of failing honeypot assignments.

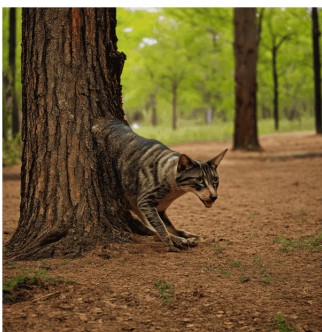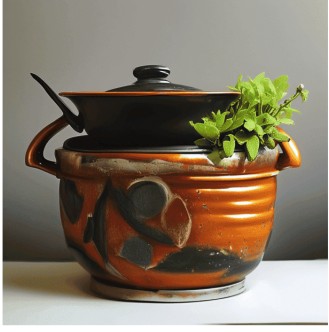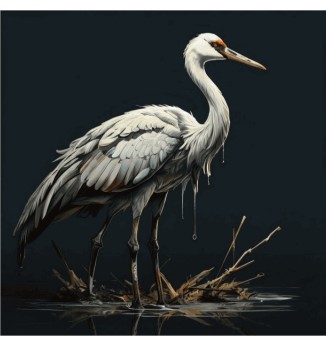

Figure 6: Two hard and one easy labeling examples. From left to right: "bark", "pot", "crane". The first two examples are considered difficult to label as duplicates since their meanings are linked implicitly through an association. The animal depicted in the first image, appearing to emerge from the tree bark, resembles a cat. Although cats are known not to bark, the cat's presence may evoke associations with a barking dog, thereby justifying the classification of the image as a duplicate. The word "pot", which is depicted in the second image, has 4 different meanings (flower pot, tea pot, saucepan, boiling pot), making it challenging to reliably identify which meanings are actually depicted in the image. The last example is easy to label since there is no second meaning present, and the image is clearly not a duplicate.

## A.2 PROMPT EXAMPLES

### A.2.1 VLLM-BASED AUTOMATIC EVALUATION

**Example of one-stage prompt for duplicate detection**

```
There is a problem, which is called as Homonym Duplication. It is when in the
image, which was generated by text2image model, there are several senses of the
input prompt. This image is generated by a neural network for a multi-senses
short prompt: ``basket''. This prompt can take several values:
1) a wicker item that serves as a storage, packing or carrying case. It is
usually a handicraft, the methods of making which vary from region to region.
2) a structure made of a metal ring with a net hanging from it for throwing the
ball. It is used for playing basketball. Possibly there are other values. The
meaning of the homonym may be implicit in the picture. For each meaning, reason
step-by-step and mark the presence in the picture and provide the answer in the
following template:
sense_1: [Explicit|Implicit|Absent], justification
sense_2: [Explicit|Implicit|Absent], justification
...
[Reasoning, summarization].
DUPLICATE: [TRUE|FALSE].
```

An image may be connected to a meaning implicitly, through association or related meaning. Therefore, be very attentive and carefully study the picture for the presence of the listed meanings, even if they are presented implicitly. Note:
* Implicit covers metaphors, hints, visual puns.
* Base judgments only on what is visible (including any text shown).
* Base your final words in answer as DUPLICATE: TRUE, or DUPLICATE: FALSE.

---

### Example of multi-stage prompt for duplicate detection

You are analyzing an image generated by a diffusion model for potential Homonym Duplication.
TASK: Determine if this image contains multiple meanings of the word ``basket''.
STEP 1 - VISUAL INVENTORY
List all objects, elements, and visual features you observe in the image.
STEP 2 - MEANING ANALYSIS
The word ``basket'' can have these meanings:
1) a wicker item that serves as a storage, packing or carrying case. It is usually a handicraft, the methods of making which vary from region to region.
2) a structure made of a metal ring with a net hanging from it for throwing the ball. It is used for playing basketball. Possibly there are other values.
For each meaning, analyze:
- EXPLICIT presence: Direct visual representation
- IMPLICIT presence: Suggested through context, associations, or related elements
- ABSENT: No connection to this meaning
Format:
meaning_X: [EXPLICIT|IMPLICIT|ABSENT] - detailed justification
STEP 3 - ASSOCIATION MAPPING
Check for indirect connections:
- Visual metaphors or symbols
- Contextual clues that suggest meanings
- Objects that relate to but don't directly represent meanings
STEP 4 - FINAL DETERMINATION
Count meanings with EXPLICIT or IMPLICIT presence.
If 2 meanings present: DUPLICATE: TRUE
If < 2 meanings present: DUPLICATE: FALSE
Provide your structured analysis following each step.

---

### Model answer example

sense_1: Absent, justification
The image does not contain any elements that explicitly or implicitly suggest a bass guitar. There are no musical instruments, strings, or any related imagery present.
sense_2: Explicit, justification
The image clearly depicts a fish, specifically one resembling a bass fish. The underwater setting, coral reefs, and the fish's features such as its fins, scales, and open mouth are all consistent with the depiction of a bass fish.
[Reasoning, summarization].
The image contains an explicit representation of a bass fish, aligning with the second sense of the word ``bass''. There is no indication of the first sense, which refers to a bass guitar. Since the image only represents one of the possible meanings of the word ``bass'', it does not exhibit homonym duplication.
DUPLICATE: FALSE.

### A.2.2 Prompt Expansion

> **Example of the prompt used to expand the original single-word prompts**
>
> ```
> You are a prompt engineer. Your mission is to expand prompts written by user.
> You should provide the best prompt for text to image generation in English in
> 1-2 sentences. You MUST INCLUDE given word in its original form in a prompt.
> Expand prompt for this word: ``bass''. Respond ONLY WITH the example of an
> expanded prompt, nothing else.
> ```

> **Example of the expanded prompt for the word "bass"**
>
> ```
> A serene lake at sunrise, with a sleek bass leaping out of the water,
> surrounded by lush greenery and mist.
> ```

### A.3 Named Entity Bias

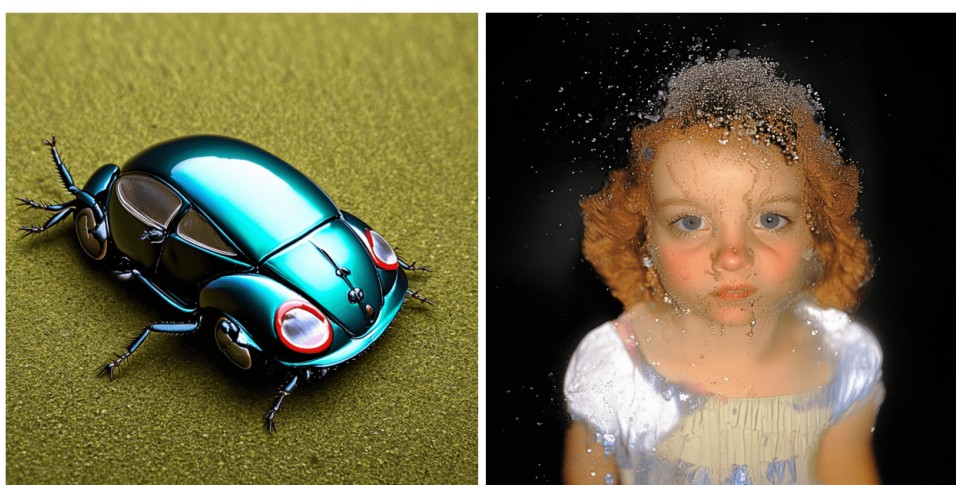

Figure 7: Examples of named entity bias. On the left, the image generated for the prompt "beetle" depicts a car resembling a Volkswagen Beetle in the form of an actual insect. On the right, the image for the prompt "jelly" shows a girl (interpreting "Jelly" as a female name) morphing into jelly.

### A.4 Additional Tables

Table 3: Distribution of homonym duplication in an image by homonyms. We present statistics on the checkboxes: no selection, one selection, and two or more selections.

| homonym | General | | | Prompt Expansion | | |
|---|---|---|---|---|---|---|
| | nothing | one | two+ | nothing | one | two+ |
| agent | 5.8 | 94.2 | 0 | 2 | 98 | 0 |
| anchor | 0.9 | 99.1 | 0 | 0 | 100 | 0 |
| angle | 61.5 | 38.5 | 0 | 86 | 14 | 0 |
| ash | 72.4 | 26 | 1.6 | 0 | 40 | 60 |
| baby | 0.2 | 99.5 | 0.3 | 0 | 100 | 0 |
| ball | 6 | 84.2 | 9.8 | 22 | 58 | 20 |
| band | 3.5 | 96.5 | 0 | 0 | 100 | 0 |
| bank | 7.1 | 92.4 | 0.5 | 0 | 100 | 0 |

Table 3: Distribution (%) of homonym duplication in an image by homonyms (continuation).

| homonym | General | | | Beautification | | |
|---|---|---|---|---|---|---|
| | nothing | one | two+ | nothing | one | two+ |
| bar | 0.4 | 99.6 | 0 | 0 | 100 | 0 |
| bark | 1.6 | 89.3 | 9.1 | 0 | 98 | 2 |
| barrel | 0.2 | 99.5 | 0.3 | 0 | 100 | 0 |
| basket | 0.7 | 91.8 | 7.5 | 0 | 100 | 0 |
| bass | 3.5 | 92.7 | 3.8 | 2 | 98 | 0 |
| batter | 31.1 | 68.9 | 0 | 8 | 92 | 0 |
| bead | 1.1 | 95.8 | 3.1 | 0 | 84 | 16 |
| beam | 28.2 | 66.2 | 5.6 | 0 | 96 | 4 |
| bed | 0 | 99.8 | 0.2 | 0 | 100 | 0 |
| bench | 1.1 | 96.7 | 2.2 | 2 | 96 | 2 |
| berth | 45.6 | 53.8 | 0.6 | 4 | 96 | 0 |
| block | 14.4 | 71.6 | 14 | 0 | 94 | 6 |
| blow | 44.2 | 51.8 | 4 | 78 | 18 | 4 |
| boil | 15.5 | 84.5 | 0 | 10 | 90 | 0 |
| bolt | 51.8 | 47.5 | 0.7 | 4 | 94 | 2 |
| bow | 18.5 | 81.1 | 0.4 | 62 | 38 | 0 |
| bowl | 0 | 99.5 | 0.5 | 0 | 100 | 0 |
| box | 6 | 93.6 | 0.4 | 2 | 96 | 2 |
| brush | 26.5 | 61.3 | 12.2 | 12 | 82 | 6 |
| buck | 0.4 | 99.6 | 0 | 0 | 100 | 0 |
| bucket | 2.4 | 96.7 | 0.9 | 0 | 100 | 0 |
| bug | 3.1 | 96.9 | 0 | 0 | 100 | 0 |
| button | 5.5 | 68.7 | 25.8 | 2 | 54 | 44 |
| cane | 18.7 | 77.8 | 3.5 | 10 | 90 | 0 |
| canvas | 6.5 | 92 | 1.5 | 26 | 74 | 0 |
| cape | 10 | 69.5 | 20.5 | 0 | 16 | 84 |
| capital | 6.5 | 51.3 | 42.2 | 0 | 100 | 0 |
| case | 66.9 | 28.2 | 4.9 | 30 | 60 | 10 |
| cell | 46.2 | 53.8 | 0 | 0 | 100 | 0 |
| charm | 30.2 | 67.8 | 2 | 48 | 52 | 0 |
| chest | 4 | 88.9 | 7.1 | 0 | 100 | 0 |
| chip | 19.5 | 77.3 | 3.2 | 14 | 86 | 0 |
| clove | 53.1 | 46.9 | 0 | 18 | 82 | 0 |
| club | 33.8 | 61.1 | 5.1 | 2 | 92 | 6 |
| coach | 36 | 63.1 | 0.9 | 8 | 90 | 2 |
| cobbler | 25.3 | 74.4 | 0.3 | 8 | 92 | 0 |
| collar | 2.5 | 68 | 29.5 | 4 | 72 | 24 |
| court | 24.2 | 74.5 | 1.3 | 36 | 62 | 2 |
| crane | 0 | 97.5 | 2.5 | 0 | 100 | 0 |
| cricket | 9.8 | 89.5 | 0.7 | 0 | 100 | 0 |
| crown | 1.6 | 98.2 | 0.2 | 0 | 100 | 0 |
| date | 44.4 | 48.7 | 6.9 | 4 | 96 | 0 |
| deck | 6.9 | 83.8 | 9.3 | 0 | 100 | 0 |
| diamond | 0.2 | 42.4 | 57.4 | 0 | 62 | 38 |
| ear | 3.6 | 96.4 | 0 | 38 | 60 | 2 |
| fan | 30.9 | 66.2 | 2.9 | 18 | 80 | 2 |
| fence | 0.4 | 99.6 | 0 | 0 | 100 | 0 |
| file | 73.6 | 26.4 | 0 | 36 | 64 | 0 |
| flask | 10.5 | 81.6 | 7.9 | 0 | 96 | 4 |
| flute | 10.4 | 89.5 | 0.1 | 22 | 78 | 0 |
| font | 26.7 | 69.5 | 3.8 | 24 | 76 | 0 |
| fork | 6.4 | 93.5 | 0.1 | 12 | 84 | 4 |
| funnel | 9.3 | 78.5 | 12.2 | 40 | 60 | 0 |

Table 3: Distribution (%) of homonym duplication in an image by homonyms (continuation).

| homonym | General | | | Beautification | | |
|---|---|---|---|---|---|---|
| | nothing | one | two+ | nothing | one | two+ |
| gate | 0.4 | 99.6 | 0 | 0 | 100 | 0 |
| ghost | 0.4 | 73.3 | 26.3 | 0 | 20 | 80 |
| gin | 13.5 | 86.5 | 0 | 0 | 100 | 0 |
| glasses | 0.5 | 98.2 | 1.3 | 2 | 98 | 0 |
| ground | 3.8 | 76.5 | 19.7 | 0 | 80 | 20 |
| gum | 27.8 | 62.5 | 9.7 | 50 | 50 | 0 |
| hatch | 47.5 | 50.4 | 2.1 | 62 | 38 | 0 |
| heel | 6 | 34.5 | 59.5 | 2 | 62 | 36 |
| horn | 6.4 | 78.9 | 14.7 | 2 | 98 | 0 |
| jam | 33.3 | 66.5 | 0.2 | 14 | 86 | 0 |
| jar | 0.2 | 98.9 | 0.9 | 0 | 100 | 0 |
| jet | 9.5 | 80.5 | 10 | 0 | 94 | 6 |
| jumper | 4.9 | 74.9 | 20.2 | 2 | 98 | 0 |
| junk | 32 | 67.6 | 0.4 | 10 | 90 | 0 |
| lace | 0.5 | 99.1 | 0.4 | 0 | 100 | 0 |
| leg | 7.8 | 77.1 | 15.1 | 0 | 96 | 4 |
| line | 33.8 | 56 | 10.2 | 30 | 48 | 22 |
| litter | 22.2 | 74.2 | 3.6 | 0 | 100 | 0 |
| lock | 3.3 | 96.7 | 0 | 2 | 98 | 0 |
| log | 21.8 | 78.2 | 0 | 0 | 100 | 0 |
| magazine | 24 | 76 | 0 | 10 | 90 | 0 |
| mail | 10.5 | 89.5 | 0 | 2 | 98 | 0 |
| match | 46.2 | 48.4 | 5.4 | 4 | 80 | 16 |
| mate | 25.6 | 74.2 | 0.2 | 8 | 92 | 0 |
| mine | 71.5 | 24.7 | 3.8 | 4 | 88 | 8 |
| mint | 17.5 | 82.4 | 0.1 | 4 | 96 | 0 |
| model | 12 | 88 | 0 | 18 | 82 | 0 |
| mold | 16.9 | 82.9 | 0.2 | 18 | 82 | 0 |
| mole | 24.9 | 66 | 9.1 | 14 | 86 | 0 |
| mouse | 0 | 98.9 | 1.1 | 0 | 100 | 0 |
| mug | 1.3 | 90.9 | 7.8 | 0 | 98 | 2 |
| nail | 6.4 | 93.5 | 0.1 | 4 | 96 | 0 |
| needle | 23.3 | 52.5 | 24.2 | 20 | 70 | 10 |
| net | 16.7 | 75.3 | 8 | 28 | 32 | 40 |
| note | 22.4 | 76.5 | 1.1 | 0 | 98 | 2 |
| notebook | 7.1 | 92 | 0.9 | 0 | 100 | 0 |
| nut | 17.6 | 81.1 | 1.3 | 0 | 100 | 0 |
| oil | 24.7 | 66.7 | 8.6 | 44 | 52 | 4 |
| organ | 16.5 | 83.1 | 0.4 | 30 | 70 | 0 |
| pack | 26.2 | 70.9 | 2.9 | 0 | 98 | 2 |
| palm | 0 | 95.3 | 4.7 | 0 | 100 | 0 |
| park | 0.2 | 97.6 | 2.2 | 0 | 100 | 0 |
| party | 0.5 | 99.5 | 0 | 0 | 100 | 0 |
| pen | 21.8 | 78.2 | 0 | 0 | 100 | 0 |
| pipe | 2.4 | 93.1 | 4.5 | 14 | 86 | 0 |
| pitcher | 6.4 | 92.7 | 0.9 | 0 | 100 | 0 |
| plane | 0 | 99.8 | 0.2 | 0 | 100 | 0 |
| plant | 0 | 100 | 0 | 0 | 100 | 0 |
| plate | 0.5 | 99.5 | 0 | 34 | 66 | 0 |
| plot | 31.5 | 67.1 | 1.4 | 16 | 84 | 0 |
| plug | 11.6 | 88.2 | 0.2 | 98 | 2 | 0 |
| pod | 61.3 | 36.7 | 2 | 8 | 92 | 0 |
| pole | 7.6 | 92.4 | 0 | 2 | 98 | 0 |

Table 3: Distribution (%) of homonym duplication in an image by homonyms (continuation).

| homonym | General | | | Beautification | | |
|---|---|---|---|---|---|---|
| | nothing | one | two+ | nothing | one | two+ |
| pool | 0.2 | 99.5 | 0.3 | 2 | 98 | 0 |
| pot | 4.7 | 82.5 | 12.8 | 0 | 96 | 4 |
| press | 54.7 | 42.5 | 2.8 | 24 | 48 | 28 |
| pump | 13.1 | 86.4 | 0.5 | 30 | 70 | 0 |
| quiver | 44.9 | 54.5 | 0.6 | 84 | 16 | 0 |
| rail | 5.3 | 90.4 | 4.3 | 0 | 98 | 2 |
| ring | 0.5 | 99.5 | 0 | 0 | 100 | 0 |
| roll | 18.9 | 78.2 | 2.9 | 62 | 38 | 0 |
| row | 31.3 | 52.2 | 16.5 | 30 | 66 | 4 |
| rug | 14.2 | 72.4 | 13.4 | 0 | 80 | 20 |
| ruler | 12.2 | 87.5 | 0.3 | 12 | 88 | 0 |
| scale | 21.8 | 69.5 | 8.7 | 14 | 86 | 0 |
| screen | 16.7 | 80.9 | 2.4 | 24 | 76 | 0 |
| seal | 9.5 | 78.4 | 12.1 | 0 | 100 | 0 |
| sewer | 2.2 | 97.8 | 0 | 2 | 98 | 0 |
| sheet | 17.3 | 78.2 | 4.5 | 18 | 82 | 0 |
| shower | 13.1 | 85.3 | 1.6 | 22 | 74 | 4 |
| sink | 6.5 | 92.5 | 1 | 2 | 98 | 0 |
| skate | 0.7 | 97.8 | 1.5 | 0 | 100 | 0 |
| skeleton | 0 | 100 | 0 | 4 | 96 | 0 |
| slough | 35.8 | 64.2 | 0 | 0 | 100 | 0 |
| sole | 40.9 | 45.8 | 13.3 | 6 | 90 | 4 |
| sow | 38.5 | 61.5 | 0 | 0 | 100 | 0 |
| space | 0.2 | 99.6 | 0.2 | 0 | 100 | 0 |
| spirit | 28.4 | 71.6 | 0 | 4 | 96 | 0 |
| spoon | 2.5 | 97.3 | 0.2 | 4 | 96 | 0 |
| spring | 0.2 | 83.6 | 16.2 | 0 | 20 | 80 |
| spur | 81.1 | 18.9 | 0 | 10 | 90 | 0 |
| square | 10.9 | 76.5 | 12.6 | 2 | 88 | 10 |
| squash | 0.4 | 99.1 | 0.5 | 4 | 96 | 0 |
| staff | 14.2 | 85.3 | 0.5 | 18 | 82 | 0 |
| stamp | 1.5 | 85.6 | 12.9 | 8 | 82 | 10 |
| store | 0.4 | 82 | 17.6 | 0 | 100 | 0 |
| straw | 5.1 | 90.2 | 4.7 | 8 | 92 | 0 |
| string | 14.5 | 82.4 | 3.1 | 68 | 30 | 2 |
| table | 0.4 | 99.6 | 0 | 0 | 100 | 0 |
| tail | 27.6 | 71.5 | 0.9 | 2 | 98 | 0 |
| tank | 0.4 | 99.1 | 0.5 | 0 | 100 | 0 |
| tear | 36.2 | 61.3 | 2.5 | 44 | 56 | 0 |
| temple | 0 | 100 | 0 | 0 | 100 | 0 |
| tick | 34.5 | 64.4 | 1.1 | 22 | 78 | 0 |
| tie | 23.5 | 76.5 | 0 | 6 | 94 | 0 |
| tip | 83.5 | 16.5 | 0 | 30 | 70 | 0 |
| toast | 6.7 | 92.7 | 0.6 | 0 | 100 | 0 |
| track | 38.5 | 51.5 | 10 | 4 | 84 | 12 |
| train | 0.4 | 99.6 | 0 | 0 | 100 | 0 |
| trunk | 12 | 72.2 | 15.8 | 0 | 88 | 12 |
| urn | 11.1 | 88.9 | 0 | 0 | 100 | 0 |
| vane | 75.1 | 22.4 | 2.5 | 0 | 88 | 12 |
| veil | 0.9 | 97.3 | 1.8 | 0 | 46 | 54 |
| vessel | 6.4 | 90.5 | 3.1 | 0 | 100 | 0 |
| washer | 10.9 | 88.5 | 0.6 | 0 | 100 | 0 |
| watch | 4 | 85.1 | 10.9 | 0 | 100 | 0 |

Table 3: Distribution (%) of homonym duplication in an image by homonyms (continuation).

| homonym | General | | | Beautification | | |
|---|---|---|---|---|---|---|
| | nothing | one | two+ | nothing | one | two+ |
| wave | 0.2 | 96.5 | 3.3 | 0 | 96 | 4 |
| whiskers | 16.4 | 82 | 1.6 | 0 | 100 | 0 |
| window | 0.2 | 99.6 | 0.2 | 0 | 100 | 0 |
| wing | 0.5 | 95.6 | 3.9 | 2 | 98 | 0 |

Table 4: Alignment between human evaluation and CLIP-based automatic evaluation. Sense representation type in a prompt differs from those described in section 5. For CLIP-like rankers, $g_1$ denotes the English sense definition, $g_2$ the Russian sense definition, and $g_3$ a short Russian translation equivalent. For each image, we obtain between two and six CLIPScore values (depending on the number of senses of the given homonym). The Factor column indicates which CLIPScore value is used to calculate the correlation with the results of human evaluation (i.e., to what extent the coefficient explains and correlates with human evaluations).

| Model | Sense representation type | Factor | $r \uparrow$ | $\rho \uparrow$ | AUROC* $\uparrow$ |
|---|---|---|---|---|---|
| OpenAI CLIP-L/14 Radford et al. (2021) | $g_1$ | top-1 | 0.054 | 0.049 | 0.565 |
| | | top-2 | 0.159 | 0.166 | 0.722 |
| | | top-1 + top-2 | 0.123 | 0.127 | 0.669 |
| | | top-2 - top-1 | 0.101 | 0.103 | 0.638 |
| | $g_2$ | top-1 | 0.023 | 0.025 | 0.534 |
| | | top-2 | 0.023 | 0.025 | 0.533 |
| | | top-1 + top-2 | 0.024 | 0.026 | 0.535 |
| | | top-2 - top-1 | 0.003 | 0.002 | 0.503 |
| mSigLIP Zhai et al. (2023) | $g_1$ | top-1 | 0.058 | 0.047 | 0.563 |
| | | top-2 | 0.192 | 0.202 | 0.769 |
| | | top-1 + top-2 | 0.146 | 0.153 | 0.705 |
| | | top-2 - top-1 | 0.126 | 0.128 | 0.670 |
| | $g_2$ | top-1 | 0.052 | 0.056 | 0.575 |
| | | top-2 | 0.125 | 0.122 | 0.663 |
| | | top-1 + top-2 | 0.098 | 0.097 | 0.629 |
| | | top-2 - top-1 | 0.081 | 0.088 | 0.617 |
| | $g_3$ | top-1 | 0.066 | 0.057 | 0.577 |
| | | top-2 | 0.153 | 0.160 | 0.713 |
| | | top-1 + top-2 | 0.126 | 0.126 | 0.668 |
| | | top-2 - top-1 | 0.101 | 0.096 | 0.627 |
| SigLIP2 Tschannen et al. (2025) | $g_1$ | top-1 | 0.043 | 0.04 | 0.553 |
| | | top-2 | 0.184 | 0.183 | 0.744 |
| | | top-1 + top-2 | 0.132 | 0.134 | 0.679 |
| | | top-2 - top-1 | 0.119 | 0.119 | 0.658 |
| | $g_2$ | top-1 | 0.061 | 0.055 | 0.573 |
| | | top-2 | 0.155 | 0.158 | 0.711 |
| | | top-1 + top-2 | 0.133 | 0.133 | 0.677 |
| | | top-2 - top-1 | 0.092 | 0.095 | 0.626 |
| | $g_3$ | top-1 | 0.049 | 0.035 | 0.546 |
| | | top-2 | 0.182 | 0.185 | 0.747 |
| | | top-1 + top-2 | 0.134 | 0.128 | 0.67 |
| | | top-2 - top-1 | 0.135 | 0.133 | 0.678 |

Table 5: Frequency of proper name occurrences. We evaluate the number of generations of different meanings across all models. It can be observed that if a homonym word has a proper name as one of its meanings, the model exhibits a pronounced bias toward generating that proper name.

| Word | Translation equivalents | | | |
|---|---|---|---|---|
| stitch | cartoon character Stitch **63.6%** | sewing stitch 18.0% | stitch in abdomen 0% | |
| bug | Volkswagen Beetle **10.9%** | sledgehammer 0% | beetle 90.2% | |
| bill | Person named Bill **69.1%** | payment bill 2.5% | banknote 23.3% | bird's bill 0.4% |
| bat | Batman **38.5%** | baseball bat 0% | bat (animal) 87.3% | |
| jelly | Person named Jelly 0.2% | gelatine dessert 93.8% | | |
| jack | Person named Jack OR an animal named Jack **95.63%** | jack fish 0% | plug 0% | perforator 0% |
| mark | Person named Mark OR an animal named Mark **58.73%** | mark on paper 2.18 % | trade mark 10.73% | |

Table 6: Distribution (%) of the number of senses per image annotated by human evaluation. Each model generated 8,550 images, ensuring that the resulting metrics are statistically reliable. Statistics for prompt expansion are reported only for the Pixart Alpha model owing to the high cost of annotation.

| | General | | | Prompt Expansion | | |
|---|---|---|---|---|---|---|
| model | nothing | one | two+ | nothing | one | two+ |
| Pixart Alpha | 29.7 | 64.8 | 5.5 | 10.7 | 84.3 | 5.0 |
| Cogview 4 | 25.3 | 71.8 | 2.9 | - | - | - |
| Flux 1 dev | 10.6 | 84.5 | 4.9 | - | - | - |
| Flux 1 schell | 9.9 | 84.1 | 6.0 | - | - | - |
| Kandinsky 3 | 11.9 | 82.0 | 6.1 | - | - | - |
| Pixart Sigma | 25.6 | 70.3 | 4.1 | - | - | - |
| Playground 2.5 | 9.1 | 83.9 | 7.0 | - | - | - |
| SD 3 Medium | 13.7 | 81.8 | 4.5 | - | - | - |
| SD 3.5 Large | 7.4 | 87.6 | 5.0 | - | - | - |
| SD 3.5 Medium | 12.3 | 83.0 | 4.7 | - | - | - |
| SD XL | 28.5 | 68.1 | 3.4 | - | - | - |

Table 7: Prompt expansion results for Russian prompts. The resulting HDR and WSR metrics are obtained via human evaluation.

| Russian word | English translation equivalent | W/o prompt expansion | | With prompt expansion | |
|---|---|---|---|---|---|
| | | WSR ↓ | HDR ↓ | WSR ↓ | HDR ↓ |
| финик | date (fruit) | 68 | 20 | 58 | 0 |
| дата | date (social meeting) | 100 | 0 | 72 | 0 |
| свидание | date (in the calendar) | 14 | 20 | 4 | 0 |
| весна | spring (season) | 0 | 46 | 0 | 30 |
| родник | spring (water) | 54 | 46 | 0 | 84 |
| пружина | spring (metal coil) | 100 | 0 | 44 | 0 |
| ноготь | nail (part of the finger) | 0 | 0 | 34 | 0 |
| гвоздь | nail (fastener) | 100 | 0 | 44 | 0 |
| таблица | table (chart) | 100 | 0 | 100 | 0 |
| стол | table (desk) | 0 | 0 | 0 | 0 |
| линейка | ruler (measuring tool) | 30 | 0 | 14 | 0 |
| правитель | ruler (leader) | 98 | 0 | 4 | 0 |
| почтовая марка | stamp (post) | 8 | 2 | 36 | 12 |
| штамп | stamp (mark) | 90 | 2 | 4 | 0 |
| тростник | cane (plant) | 94 | 0 | 0 | 0 |
| трость | cane (walking aid) | 68 | 0 | 34 | 2 |
| пепел | ash (powder left after burning) | 44 | 4 | 8 | 44 |
| ясень | ash (tree) | 90 | 4 | 2 | 0 |
| мята | mint (plant) | 0 | 0 | 0 | 0 |
| монетный двор | mint (coin factory) | 100 | 0 | 58 | 0 |
| дуло | barrel (of a gun) | 100 | 0 | 48 | 4 |
| бочка | barrel (container) | 0 | 0 | 0 | 0 |
| масло | oil (for cooking) | 28 | 6 | 12 | 0 |
| нефть | oil (petroleum) | 84 | 6 | 20 | 2 |
| столица | capital (metropolis) | 18 | 50 | 0 | 2 |
| капитал | capital (money and possesions) | 100 | 0 | 92 | 8 |
| капитель | capital (part of the pillar) | 36 | 50 | 18 | 0 |
| джемпер | jumper (clothing) | 24 | 76 | 12 | 0 |
| прыгун | jumper (someone who jumps) | 0 | 76 | 4 | 4 |
| ромб | diamond (rhombus) | 20 | 80 | 4 | 88 |
| алмаз | diamond (stone) | 0 | 80 | 10 | 20 |
| джонка | junk (vessel) | 100 | 0 | 0 | 0 |
| барахло | junk (trash) | 4 | 0 | 0 | 0 |
| пальма | palm (tree) | 0 | 20 | 0 | 0 |
| ладонь | palm (part of the hand) | 80 | 20 | 58 | 28 |
| крикет | cricket (sport game) | 78 | 2 | 16 | 0 |
| сверчок | cricket (insect) | 20 | 2 | 8 | 0 |

