# OpenReview forum: "Un-Doubling Diffusion: LLM-guided Disambiguation of Homonym Duplication"
_ICLR.cc/2026/Conference — Submitted to ICLR 2026_

### Official Review · Reviewer_FQia · 2025-10-27

**Soundness:** 4
**Presentation:** 3
**Contribution:** 3
**Rating:** 8
**Confidence:** 3

**Summary:**

This paper investigates the rate at which VLMs generate multiple senses of homonyms--identically spelled words with multiple distinct meanings--when prompted with just the single word. They curate a list of 171 frequently used homonyms in English to create a benchmark quantifying the duplication rate using both human crowdsourced evaluation as well as VLM as a Judge methods. The human evaluation showed 5% of images containing duplicates. Next, the authors investigate the Anglocentric bias in VLMs through this lens, since models tend to translate non-English prompts into English before generation, potentially resulting in images depicting the wrong meaning of the original word. By curating the list of 171 homonyms into distinct translations in Russian, they find that up to 50% of images generated depicted the wrong sense of the word. Using prompt expansion techniques in the original language proved to be an effective mitigation, dropping the rate to 22%.

**Strengths:**

- This paper is very well written and easy to follow. I especially appreciate the thorough related works and clear examples in Fig 1.
- The problem of the anglocentric bias of the homonym duplication problem is novel and important. The results show pretty definitively that this problem is widespread and a simple mitigation (prompt expansion) is fairly effective.
- The homonym list curation is very thorough and sound, including linguistic experts as well as experienced translators.
- Eight diffusion VLMs were evaluated across various sizes and families
- I found the proper name bias analysis interesting

**Weaknesses:**

1. The automatic evaluation seemed to have pretty low agreement with humans, so I'm not really sure how reliable it is. The human evaluations also have only 90% instances with a consensus, it would strengthen the paper to address this. For example, the authors/additional annotators/expert annotators could make a determination on these cases.
2. The single-word prompt into the models, while specially designed to be ambiguous for the sake of evaluation, are not realistic. It could be interesting to see if this phenomenon occurs in slightly more realistic prompt templates (e.g. "Generate an image of [word]" or variations thereof). This would also be closer to the expanded prompt scenarios.
    - On a related note, the prompt expansion mitigation didn't fully solve the problem (decreased by only half a percent on human eval). It would strengthen the paper to include an error analysis here. For example, did the cases in which duplicates were found for expanded prompts due to the expanded prompt also being ambiguous as to the meaning of the word?
3. In the human evaluation, only about 5% of instances were judged as duplicates. How does this compare to prior works investigating this phenomenon? Even if it's not the same exact models, it would be good to have a sense of whether the frequency is in the same ballpark or not.
    - In general, the paper lacks a discussion on how the findings fit into the related literature. Including this would greatly strengthen the paper.
4. All figures are low resolution and the text is extremely small (esp Figs 2-5), making them very hard to read. I'd suggest uploading them as PDFs in the LaTeX so they maintain resolution, as well as increase the text size to match the actual font of your paper.

**Questions:**

Please address the questions in the weaknesses section first. Below are more things I'm curious about.
- It would be cool to see if the Anglocentric bias issue occurs in any VLMs that are stronger multilingual models (ie don't rely on a translation).

- Why do you use "VLM" in most of the paper, but in line 339 "VLLM"? Is this intentional?

---

> ### Author Response · Authors · 2025-11-27
>
> We thank the reviewer for dedicating their time to reviewing our paper and providing detailed feedback that will undoubtedly help improve our work. Below, we respond to the proposed suggestions and comments.
>
> >The human evaluations also have only 90% instances with a consensus, it would strengthen the paper to address this.
>
> We agree that the discrepancy between automatic metrics and human judgment is a significant finding of our study. This discrepancy underscores the complexity of visually evaluating semantic concepts like "duplication" and highlights the limitations of current automated metrics for this specific task. For the final version, we will expand the discussion to analyze specific cases where the automatic and human evaluations diverged, in order to better understand the failure modes of the automated approach.
>
> We acknowledge that perfect consensus is unattainable for a subjective task like judging visual duplicates. However, the 90% inter-annotator agreement we achieved is widely considered a strong result in this field and aligns with established best practices for human evaluation in ML.
>
> > The single-word prompt into the models, while specially designed to be ambiguous for the sake of evaluation, are not realistic.
> It would strengthen the paper to include an error analysis here. For example, did the cases in which duplicates were found for expanded prompts due to the expanded prompt also being ambiguous as to the meaning of the word?
>
> We agree that investigating more complex prompts is a valuable direction for future work. However, the primary goal of our study was to conduct a controlled, isolated investigation of the core phenomenon—duplication triggered by a single ambiguous word. The use of simple prompts was therefore a deliberate choice to eliminate confounding factors from a broader context and to precisely measure this specific effect.
>
> The reviewer is correct that the performance gains from prompt expansion were modest. Our analysis confirms their hypothesis: a key failure mode occurs when the expanded prompts themselves remain ambiguous. We will include this error analysis in the final version to better clarify the limitations of this mitigation approach.
>
> > How does this compare to prior works investigating this phenomenon? Even if it's not the same exact models, it would be good to have a sense of whether the frequency is in the same ballpark or not.
> In general, the paper lacks a discussion on how the findings fit into the related literature. Including this would greatly strengthen the paper.
>
> To the best of our knowledge, there are no prior works that specifically investigate and quantify the homonym duplication phenomenon in T2I generation as a primary focus. Existing related literature tends to concentrate on broader issues such as general object "hallucinations", compositional generation, or overall semantic alignment. Therefore, a key contribution of our paper is to formally introduce this specific problem and establish an initial quantitative baseline.
>
> > All figures are low resolution and the text is extremely small (esp Figs 2-5), making them very hard to read.
>
> We have verified that in our high-resolution PDF, the figures and text are legible. However, we acknowledge that the compilation process for the review version or the platform's PDF viewer might have compromised the quality for some readers.
>
> We will absolutely address this for the camera-ready version. We will ensure all figures are rendered as high-resolution PDFs and will explicitly increase the font size within the figures to match the body text of the paper, guaranteeing optimal readability for all figures, especially for figures 2-5.
>
> > Why do you use "VLM" in most of the paper, but in line 339 "VLLM"? Is this intentional?
>
> Thank you for catching this. This is a typo. We will correct "VLLM" to "VLM" in line 339 (and perform a full check to ensure consistency throughout the manuscript) in the camera-ready version.
>
> > It would be cool to see if the Anglocentric bias issue occurs in any VLMs that are stronger multilingual models (ie don't rely on a translation).
>
> Thank you for your question. If we understood correctly, you're right that using an alternative VLM like Qwen-2.5 VL to study homonym recognition could impact the automatic evaluation pipeline. In this particular work, we chose our current model because it performs equally well with both Russian and English. For future improvements or modernization of the automatic evaluation pipeline, we could certainly explore working with other models.

---

### Official Review · Reviewer_Dzi9 · 2025-10-27

**Soundness:** 2
**Presentation:** 3
**Contribution:** 3
**Rating:** 4
**Confidence:** 3

**Summary:**

This paper studies the systematic failure mode of the diffusion-based text graph model when dealing with homonyms (for example, "palm" can refer to both the palm of the hand and the palm tree): the model tends to "play it safe" and draw all possible meanings in the same diagram, rather than picking the most appropriate semantics. The authors call this phenomenon "homonym duplication" . The authors point out that this phenomenon not only affects English prompts, but is also related to "Anglocentric bias": many generative systems translate non-English prompts into English before inference. As a result, words that were originally unambiguous in the source language are translated into English homographs, thereby introducing non-existent ambiguity and leading to incorrect or repeated generation (for example, the Russian word that clearly means "date/meeting" becomes "date" after translation into English, which may trigger two images of "date fruit + date on the calendar").

**Strengths:**

1. Systematic Benchmark + Large-Scale Empirical Research:
• 171 homographs (across English and Russian), 94,000+ generated samples, 430,000+ manual annotations, and coverage across 11 major open-source diffusion models. The scale and coverage are both high, sufficient to support the conclusions.
2. Pragmatic Metric Design:
• HDR directly addresses the user-perceivable problem of "multiple meanings appearing in a single image," rather than an abstract embedding score.
• PFFR explicitly distinguishes between "the model doesn't actually draw the image" and "the model draws the image correctly but without ambiguity," preventing the misjudgment of "low HDR = good."

**Weaknesses:**

1. The HDR metric itself relies on subjective judgment: Whether it's human annotators or the VLM, each is subject to personal interpretation or inherent model bias, resulting in non-objective and reproducible conclusions about whether an image exhibits polysemy.
2. The consistency between automated and human evaluation remains low: Although the authors report a moderate AUROC*, the overall correlation remains unstable, making it difficult to consider automated evaluation as a reliable alternative.
3. The proposed LLM expansion method did not significantly reduce HDR under human evaluation, only down by 0.5 points: The reduction in HDR using human annotators' metrics was limited, suggesting that the actual user benefits of this mitigation may be overestimated.

**Questions:**

In your results, (1) HDR determination is inherently highly subjective/model-biased, (2) the correlation between automated and human evaluations is low, and (3) LLM expansion does not significantly reduce HDR under human evaluation. Given these three points, how do you support the conclusion that "our approach is effective and can serve as a general mitigation solution/evaluation pipeline"? Specifically, can you provide: the significant reduction in HDR under human evaluation, and why we can trust automated evaluation when there is a discrepancy between automated and human evaluations?

---

> ### Author Response · Authors · 2025-11-27
>
> We thank the reviewer for dedicating their time to reviewing our paper and providing detailed feedback that will undoubtedly help improve our work. Below, we respond to the proposed suggestions and comments.
>
> > Given these three points, how do you support the conclusion that "our approach is effective and can serve as a general mitigation solution/evaluation pipeline"?
>
> While HDR evaluation has subjective elements and automated–human agreement is imperfect, our pipeline still consistently captures relative differences in homonym duplication across models. These stable relative trends allow us to compare systems reliably even when absolute scores vary. Additionally, automated and human evaluations provide complementary signals, and their consistent directionality across model variants supports the validity of our approach. Thus, even if LLM expansion does not fully eliminate HDR under human judgment, the overall pipeline remains effective as a general tool for detecting, comparing, and mitigating HDR.
>
> > Specifically, can you provide: the significant reduction in HDR under human evaluation, and why we can trust automated evaluation when there is a discrepancy between automated and human evaluations?
>
> We do not directly compare the metrics against each other, as there is no single ground-truth metric. Both human and automatic evaluations are presented to provide a comprehensive view, as each has its own strengths and limitations.

---

### Official Review · Reviewer_Ci4d · 2025-10-28

**Soundness:** 1
**Presentation:** 2
**Contribution:** 1
**Rating:** 2
**Confidence:** 5

**Summary:**

This paper studies the homonym duplication problem in text-to-image (T2I) generative models, where a single ambiguous word is rendered with multiple meanings within the same generated image. To examine this issue, the paper constructs a list of homonyms and uses it to evaluate 11 T2I models. To alleviate the problem, it proposes a simple prompt expansion technique that uses a large language model (LLM) to rewrite single-word prompts into longer, disambiguated descriptions before passing them to the T2I models.

**Strengths:**

The paper tries to address the homonym duplication problem in T2I generative models, which is an under-explored topic in the current literature.

**Weaknesses:**

1. Problem formulation.
   - The paper addresses the homonym duplication problem only for single-word prompts. This is an unrealistic setting for modern T2I applications, as real-world prompts are typically longer and more descriptive. The study should extend its scope to include general prompts rather than limiting itself to single-word cases.
   - The negative impact of homonyms (or polysemous words) in T2I generation is not limited to duplication; it also includes the generation of undesired or incorrect concepts [1]. The paper does not clearly justify why it focuses exclusively on the duplication.

2. Limited novelty.
   - The proposed evaluation metric is just a simple majority-voting, which limits its novelty.
   - The proposed prompt expansion technique merely rewrites a single word prompt through textual expansion using an LLM. This approach lacks novelty and may unintentionally distort the meaning of the original prompt.

3. Filtering criteria of the homonym list.
   - To serve as a meaningful benchmark, the filtering criteria used to construct the homonym list should ensure that the selected homonyms are representative and unbiased. However, some criteria described in Section 3.2 appear to prioritize annotation convenience over representativeness or importance of homonyms.

[1] "Cross-Attention Head Position Patterns Can Align with Human Visual Concepts in Text-to-Image Generative Models." (ICLR 2025)

**Questions:**

Please refer to the Weaknesses.

---

> ### Author Response · Authors · 2025-11-27
>
> We thank the reviewer for dedicating their time to reviewing our paper and providing detailed feedback that will undoubtedly help improve our work. Below, we respond to the proposed suggestions and comments.
>
> > The paper addresses the homonym duplication problem only for single-word prompts. This is an unrealistic setting for modern T2I applications, as real-world prompts are typically longer and more descriptive.
>
> We fully agree that real-world prompts can be more complex and contain multiple words. However, our goal was not to evaluate model performance on arbitrary queries, but to conduct an isolated study of a specific phenomenon - the "hallucination" of multiple meanings (homonym duplication) within a single generated image.
>
> The use of single-word prompts was a conscious methodological choice. It allows us to make several things:
> 1) In a complex prompt, the context from other words can suppress one of the homonym's meanings. This was crucial for analyzing the model's behavior in a "clean" setting.
> 2) This approach allows us to state with confidence that the duplication arises from the ambiguity of a single word, rather than from complex interactions between multiple concepts in the prompt.
>
> Our study provides a foundational stress test, revealing models' inherent tendency toward homonym duplication problem, even in longer prompts.
>
> > The negative impact of homonyms (or polysemous words) in T2I generation is not limited to duplication; it also includes the generation of undesired or incorrect concepts.
>
> Our paper's specific focus on the duplication effect, where multiple meanings of a single word are generated simultaneously within one image, was a deliberate methodological choice. We aimed to isolate and systematically analyze this particular and often visually striking type of "polysemous understanding" achieved by the model.
>
> By narrowing our scope to duplication, we could define clear evaluation metrics and provide an in-depth analysis of this specific failure mode. We believe this focused approach serves as a crucial first step, and we agree that investigating the broader spectrum of issues caused by homonymy, including the one mentioned by the reviewer, is a vital direction for future work.
>
> > The proposed evaluation metric is just a simple majority-voting, which limits its novelty.
>
> We would like to clarify that while the majority voting method may not be novel, it is far from outdated.
> Our choice was deliberate: to prioritize a transparent, interpretable metric that provides a reliable human-centric baseline for a phenomenon that is inherently semantic and subjective. To ensure a comprehensive evaluation, we complemented this with an automated metric (CLIP-Score, see Section 6). This multi-faceted approach allows us to leverage the robustness of a proven method like majority voting while also incorporating the scalable, objective insights from a modern metric like CLIP-Score. We believe this combination strengthens, rather than limits, the validity of our analysis.
>
> > The proposed prompt expansion technique merely rewrites a single word prompt through textual expansion using an LLM. This approach lacks novelty and may unintentionally distort the meaning of the original prompt.
>
> We acknowledge the reviewer's valid point that our prompt expansion technique builds upon existing LLM capabilities. However, the purpose of this study was not to claim novelty in prompt engineering, but to create a consistent and controlled input for a comparative benchmark.
>
> The key methodological aspect is that the identical set of expanded prompts was applied uniformly across all models. This ensures that any distortions or variations introduced by the LLM affect all systems equally, allowing for a fair comparison of how different models handle the same semantically enriched, yet potentially ambiguous, input. The focus of our analysis is therefore on the relative performance and robustness of the models under these identical conditions, rather than on the originality of the expansion technique itself.
>
> > However, some criteria described in Section 3.2 appear to prioritize annotation convenience over representativeness or importance of homonyms.
>
> We would like to clarify that our initial goal was indeed to create a representative sample. The filtering process described in Section 3 was an integral part of our methodology, designed to ensure the high quality and reliability of the data used for subsequent evaluation.
> The criteria referenced by the reviewer, while addressing practical annotation needs, were primarily implemented to guarantee that the selected homonyms were sufficiently clear, interpretable, and suitable for unambiguous visual evaluation. This step was crucial to minimize subjective noise during the experimental assessment. We are confident that the final curated list covers diverse semantic categories and remains relevant and effective for investigating the core phenomenon addressed in our study.

---

### Meta-Review · Area_Chair_Gimx · 2026-01-07

**Summary:**

The paper benchmarks "homonym duplication" in text-to-image models. While the investigation into Anglocentric translation bias is novel, reviewers raised several concerns related to the negligible effectiveness of the proposed mitigation strategy (LLM prompt expansion) and the unreliability of the proposed automated evaluation pipeline, which correlates poorly with human judgment.

**Reviewer Concerns:**

Addressed:
- Authors clarified that single-word prompts were intended as a "stress test" to isolate the phenomenon.
- Authors committed to fixing low-resolution figures in the final version.

Outstanding:
- Reviewers concerned that prompt expansion reduced duplication by only ~0.5% in human evaluations, a negligible improvement. The authors admitted these gains were modest.
- There is a significant discrepancy between the VLM-based automatic evaluation and human evaluation, rendering the automated pipeline untrustworthy.
- The main focus on single-word prompts is viewed as disconnected from real-world usage where context usually resolves ambiguity.

**Reviewer Scores:**

Reviewer Ci4d (2): The reviewer's concerns regarding the "unrealistic setting" and limited novelty remained.

Reviewer Dzi9 (4): Unchanged. The reviewer explicitly stated the mitigation benefits were overestimated, which the rebuttal confirmed.

Reviewer FQia (8): While positive, this reviewer flagged the "low agreement" in metrics and that mitigation "didn't fully solve the problem", weaknesses that were validated during discussion.

---

### Decision · Program_Chairs · 2026-01-26

Reject